# Risk Factors and Outcomes for Multidrug Resistant *Pseudomonas aeruginosa* Infection in Immunocompromised Patients

**DOI:** 10.3390/antibiotics11111459

**Published:** 2022-10-23

**Authors:** Pilar Hernández-Jiménez, Francisco López-Medrano, Mario Fernández-Ruiz, J. Tiago Silva, Laura Corbella, Rafael San-Juan, Manuel Lizasoain, Jazmín Díaz-Regañón, Esther Viedma, José María Aguado

**Affiliations:** 1Unit of Infectious Diseases, Hospital Universitario “12 de Octubre”, Instituto de Investigación Sanitaria Hospital “12 de Octubre” (imas12), 28041 Madrid, Spain; 2Department of Medicine, School of Medicine, Universidad Complutense, 28040 Madrid, Spain; 3Centro de Investigación Biomédica en Red de Enfermedades Infecciosas (CIBERINFEC), Instituto de Salud Carlos III, Madrid 28029, Spain; 4Medical Department, Merk Sharp & Dohme, 28027 Madrid, Spain; 5Department of Microbiology, Hospital Universitario “12 de Octubre”, Instituto de Investigación Sanitaria Hospital “12 de Octubre” (imas12), 28041 Madrid, Spain

**Keywords:** *Pseudomonas aeruginosa*, multi-drug resistance, immunocompromised

## Abstract

Background: *Pseudomonas aeruginosa* (PSA) infection often occurs in immunocompromised patients, which also face an increased risk of multidrug-resistant (MDR) bacteria. A deeper knowledge of the risk factors for MDR-PSA infection in this patient population may help to choose appropriate empirical antibiotic therapy. Methods: a single-center case-control (1:2) retrospective study that included 48 patients with underlying immunosuppression developing MDR-PSA infection (cases) and 96 patients also immunocompromised that were infected with non-MDR-PSA (controls) was conducted. Both groups were matched by site of infection, clinical features and type of immunosuppression. Risk factors for MDR-PSA were assessed by logistic regression. Clinical outcomes were also compared between both groups. Results: immunosuppression was due to solid cancer in 63 (43.8%) patients, solid organ transplantation in 39 (27.1%), hematological disease in 35 (24.3%) and other causes in 7 (4.9%). Independent risk factors for MDR-PSA infection were diabetes mellitus (odds ratio [OR]: 4.74; 95% confidence interval [CI]: 1.63–13.79; *p* = 0.004), antibiotic therapy in the previous 3 months (OR: 5.32; 95% CI: 1.93–14.73; *p* = 0.001), previous MDR-PSA colonization (OR: 42.1; 95% CI: 4.49–394.8; *p* = 0.001) and septic shock (OR: 3.73; 95% CI: 1.36–10.21; *p* = 0.010). MDR-PSA cases were less likely to receive adequate empirical therapy (14 [29.2%] vs. 69 [71.9%]; *p* < 0.001). 30-day clinical improvement was less common in MDR-PSA cases (25 [52.1%] vs. 76 [79.2%]; *p* = 0.001). Conclusions: diabetes mellitus, previous MDR-PSA colonization, prior receipt of antibiotics and septic shock acted as risk factors for developing MDR-PSA infections in immunocompromised patients, who have a poorer outcome than those infected with non-MDR-PSA strains.

## 1. Introduction

Despite efforts made to reduce the impact of Gram-negative bacteria (GNB) on immunosuppressed patients (e.g., prophylaxis with quinolone in case of neutropenia), this type of infection continues to be a major challenge to the clinician. In addition, the widespread use of antibiotic prophylaxis and empirical therapy with activity against GNB has contributed to the parallel increase in the rate of multidrug resistance (MDR) [1]. The development of *Pseudomonas aeruginosa* (PSA) infections is usually related to patient weakness, hospital acquisition, and use of indwelling devices and other invasive procedures [2]. These factors are particularly relevant among immunocompromised patients due to their underlying conditions and frequent contact with healthcare system [3]. For instance, high mortality rates have been reported for PSA infections in cancer patients, particularly when MDR strains are involved [4]. On the other hand, the common receipt of broad-spectrum antibiotics during episodes of febrile neutropenia and other indications of empirical therapy, prolonged antibiotic prophylaxis and invasive procedures are well-established factors contributing to the colonization by MDR-GNB. The higher prevalence of MDR-GNB infection in this patient population, in turn, worsens the prognosis and makes difficult the treatment, which often implies the use of second-line agents with higher toxicity (e.g., polymyxins) [5].

The optimal management of MDR-PSA infection in the immunocompromised host remains poorly defined, due in part to the relative scarcity of dedicated studies with a well-defined comparator group. Determining the extent to which outcomes are impacted by the MDR phenotype of the PSA isolate or the adequacy of therapy is not straightforward in patients with severe underling conditions and impaired immune response [6]. Therefore, the improvement of therapeutic approaches to MDR-PSA in immunocompromised patients has become a research priority.

The present study was conducted to assess the clinical characteristics, therapeutic management and outcomes in a single-center cohort of immunocompromised patients with MDR-PSA infection. In addition, in order to minimize potential confounding due to baseline conditions, current study analyzed the risk factors for MDR-PSA infection by means of a matched cohort composed of immunocompromised patients that also developed infection due to a PSA isolate with non-MDR phenotype.

## 2. Material and Methods

### 2.1. Study Design and Setting

This study was conducted at the University Hospital “12 de Octubre”, a 1368-bed acute-care institution in Madrid, Spain, which serves as a reference center for critically ill and oncohematological patients in a large population (445,000 in 2018) and has active solid organ transplantation (SOT) and hematopoietic stem cell transplantation (HSCT) programs. A retrospective case-control study (1:2 ratio) that included 300 patients admitted to the hospital between January 2012 and December 2017 with MDR-PSA (cases) and non-MDR-PSA infection (controls) was performed [7]. All the patients were followed-up for a minimum of 30 days. The protocol was approved by the local Clinical Research Ethics Committee, as required by the Spanish legislation for single-center retrospective studies.

For the present sub-study, we selected 48 cases and their corresponding 96 controls belonging to one of the following immunocompromised populations: SOT, solid or hematological malignancy under active chemotherapy, and other forms of immunosuppression (rheumatic diseases under immunosuppressive treatment, advanced human immunodeficiency virus [HIV] infection, and other severe primary or secondary immunodeficiencies). The presence of severe neutropenia in the febrile episode (absolute neutrophil count <500 cells/mm^3^) was recorded as a separate variable. Patients treated with ceftazidime-avibactam or ceftolozane-tazobactam were excluded due to the limited availability of these agents during the study period.

### 2.2. Matching Criteria

MDR-PSA cases and non-MDR-PSA controls were matched by the type of infection according to three criteria: site of infection, site-specific factors, and patient-specific factors. Site of infection was categorized using the criteria proposed by the Centers for Disease Control and Prevention [8] as follows: pneumonia, upper urinary tract infection (UTI), intra-abdominal infection, skin and soft tissue infection (SSTI), central venous catheter (CVC)-related bacteremia, and bacteremia of unknown origin (i.e., primary bacteremia). Site-specific factors were previous invasive procedures (e.g., catheter replacement or urinary/biliary derivation), presence of indwelling urinary catheter, and presence of other types of devices (CVC, orotracheal tube, or biliary prosthesis). Patient-specific factors recorded were related to underlying comorbidities and causes of immunosuppression.

### 2.3. Study Variables and Definitions

Basic demographics; Charlson Comorbidity index (CCI) [9] and McCabe-Jackson score [10]; major comorbidities (diabetes mellitus, cardiovascular, respiratory, liver [Child-Turcotte-Pugh class B-C cirrhosis] and renal disease); type of immunosuppression; infection severity (assessed by the Pitt’s bacteremia score [11] and development of sepsis [i.e., life-threatening organ dysfunction caused by a dysregulated host response to infection] or septic shock [sepsis with persisting hypotension requiring vasopressors to maintain mean arterial pressure ≥65 mmHg and/or serum lactate level >2 mmol/L despite adequate volume resuscitation] at the time of blood culture collection); presence of indwelling devices in place; length of hospitalization before the episode of MDR-PSA infection; results of antimicrobial susceptibility testing (AST); and empirical and targeted treatment administered (type of agent, use of combination therapy, use of extended-infusion dosing and duration); and outcomes were obtained from electronic medical records and laboratory databases by means of an standardized case report form.

Hospital and intensive care unit (ICU) admission and surgical intervention within the previous 30 days were also recorded, as well as the receipt of antibiotic therapy or history of colonization or infection with MDR-PSA or other MDR-GNB, extended-spectrum β--lactamase-producing Enterobacterales or methicillin-resistant *Staphylococcus aureus* in the preceding 3 months. Hospital-acquired infection was defined as that developed beyond 48 h from admission. History of previous MDR colonization was assessed in both surveillance samples (inguinal, rectal or nasal swabs) or clinical samples.

Complications recorded comprised the development of breakthrough (i.e., persistent bacteremia beyond 10 days after the initiation of appropriate therapy) or relapsing bacteremia (within 60 days from the end of therapy), requirement for invasive therapeutic procedures (e.g., surgical or radiologically guided drainage of collections), *Clostridioides difficile* infection, digestive tract perforation, septic thrombophlebitis, pneumonia, empyema, need for catheter removal, secondary abscesses, thromboembolic or hemorrhagic events, invasive fungal infections, drug-related toxicity, and ICU admission during the index hospitalization.

Outcomes including clinical improvement (defined by the resolution of all infection-attributable symptoms and signs by the end of treatment), probability of being discharged alive by day 30, and length of hospital stay (defined as the time interval in days between the calendar date of the positive index culture and the time of hospital discharge), were analyzed. Microbiological cure could not be assessed due to inconsistent collection of follow-up cultures.

Multidrug resistance was defined as the presence of non-susceptibility (i.e., resistance) to at least one agent in ≥3 antimicrobial categories (aminoglycosides; antipseudomonal carbapenems, cephalosporins and penicillin/ β-lactamase inhibitors; monobactams; fluoroquinolones; and polymyxins). The presence of intrinsic resistance was not taken into account to categorize a given isolate as MDR [12]. Adequate empirical antibiotic treatment was considered if at least one agent to which the isolate showed susceptibility in vitro was administered (at the appropriate dose and frequency) within the first 24 h after the sampling of the index culture. Adequate targeted therapy was considered according to the AST results once available. Combination therapy required that at least two active agents were concurrently administered for ≥24 h.

### 2.4. Statistical Analysis

Quantitative data were shown as the mean ± standard deviation (SD) or the median with interquartile ranges (IQR). Qualitative variables were expressed as absolute and relative frequencies. The Student’s t-test was used for normally distributed variables, whereas the Wilcoxon rank-sum test was applied to those with a non-normal distribution. Categorical variables were compared with the chi^2^ test or Fisher’s exact, as required. Logistic regression models were constructed to explore independent factors associated to MDR-PSA infection. Univariate analyses were separately performed for each risk factor to ascertain the corresponding odds ratios (ORs) and 95% confidence intervals (CIs). Variables with a *p* value values ≤0.2 at the univariate level were included in the regression models. The most parsimonious model was selected. All statistical analyses were performed using the Stata Statistical Software (StataCorp LLC. 2017: Release 15. College Station, TX, USA).

## 3. Results

### 3.1. Study Cohort

Overall, 144 immunocompromised patients were included (48 MDR-PSA cases and 96 non-MDR-PSA controls), whose demographics and clinical characteristics are detailed in Table 1 and Table 2. There were no significant differences between cases and controls in terms of patient age or comorbidity burden as assessed by the median CCI. Most patients had been admitted from home, although the index episode of infection was more likely to be hospital-acquired in MDR-PSA cases than in non-MDR-PSA controls (27 [40.9%] vs. 21 [26.9%]; *p* = 0.07). Cases also had a non-significant trend towards a longer hospital stay until the onset of infection (17.3 ± 34.5 vs. 9.8 ± 21.2 days; *p* = 0.11).

Causes of immunosuppression included solid cancer (63/144 patients [43.8%]), SOT (39/144 [27.1%]), hematological malignancy (35/144 [24.3%]), and other conditions (7/144 [4.9%]) (Table 1). The MDR-PSA group had higher frequency of prior colonization or infection with MDR bacteria (18 [37.5%] vs. 12 [12.5%]; *p* < 0.001). In detail, statistical significance was restricted to the previous isolation of MDR-PSA (16 [33.3%] vs. 2 [2.1%]; *p* < 0.001). Surveillance samples were more commonly obtained among MDR-PSA cases (26 [54.2%] vs. 35 [36.5%]; *p* = 0.043). To assess the potential selection bias resulting from between-group differences in the frequency of active surveillance for MDR colonization, we performed a sensitivity analysis restricted to patients with previous screening cultures. No significant differences were found in the prevalence of MDR colonization data between cases and controls in this subanalysis (54.6% vs. 45.5%; *p* = 0.16).

Hospital admission within the preceding 30 days was recorded in 18 (37.5%) MDR-PSA cases and 26 (27.1%) controls (*p*= 0.201). There were no significant differences regarding previous ICU stay or surgical intervention either. However, the receipt of antibiotic therapy in the previous 3 months was significantly more common for MDR-PSA cases (39 [81.3%] vs. 41 [42.7%]; *p* < 0.001) (Table 1).

Types of infection comprised upper UTI (75 patients [52.1%]), febrile neutropenia (either in form of neutropenic enterocolitis or without an apparent source) (21 [14.69%]), non-ventilator-associated pneumonia (12 [8.39%]), primary bacteremia (12 [8.39%]), biliary tract infection (9 [6.29%]), intraabdominal infection (6 [4.2%]), CVC-related bacteremia (3 [2.1%]), ventilator-associated pneumonia (3 [2.1%]), and SSTI (3 [2.1%]). About one-quarter of patients developed bacteremia, with no significant differences between cases and controls (*p* = 0.894). Therefore, matching for site of infection was well balanced in both groups (Table 2).

There were no differences between regarding the presence of indwelling catheters (27 [56.3%] vs. 46 [47.9%] for MDR-PSA cases and non-MDR-PSA controls, respectively; *p* = 0.346). In this case, 41 (55%) patients had a urinary catheter in place at the onset of infection, and the infection was related to urinary tract manipulation in 23 (56%) patients (4 [16%] cases vs. 19 [38%] controls; *p* = 0.051) (Table 2).

### 3.2. Results of AST

MDR-PSA isolates shown in vitro resistance to carbapenems in 37 (77.1%) cases, including an extensively drug-resistant (XDR) phenotype in 6 (12.5%). According to the major classes of antipseudomonal agents (Appendix A), 39 (81.3%) MDR-PSA isolates were susceptible to aztreonam, 22 (45.8%) to amikacin, 11 (22.9%) to meropenem, 7 (14.6%) to ceftazidime, and 5 (10.4%) to cefepime. In vitro susceptibility rates to quinolones, piperacillin-tazobactam and imipenem were below 7%, whereas all isolates remained susceptible to colistin.

### 3.3. Clinical Presentation and Therapeutic Management

Clinical presentation as sepsis was common in both MDR-PSA cases and non-MDR-PSA controls (44 [91.7%] vs. 92 [85.4%]; *p* = 0.285), although septic shock was more commonly observed in the former group (19 [39.6%] vs. 16 [16.7%]; *p* = 0.003). There were no differences in the Pitt’s bacteremia score (Table 2).

Empirical therapy comprising at least one agent with potential antipseudomonal activity was administered in 37 (77.1%) cases and 72 (75.0%) controls (*p* = 0.784). Adequate empirical antibiotic treatment according to the AST results, however, was less likely in the MDR-PSA group (14 [29.2%] vs. 69 [71.9%]; *p* < 0.001). No differences were observed between MDR-PSA and non-MDR-PSA groups in the use of combination therapy (21 [43.8%] vs. 33 [34.3%]; *p* = 0.316) or extended infusion (8 [16.7%] vs. 9 [9.4%]; *p* = 0.218).

Targeted therapy was deemed to be adequate in 38 (79.2%) MDR-PSA cases and 85 (88.5%) controls (*p* = 0.133). As expected, colistin-based regimens were more commonly used in the MDR-PSA group (15 [31.3%] vs. 1 [1.0%]; *p* < 0.001. Targeted combination therapy (17 [35.4%] vs. 18 [18.8%]; *p* = 0.028) and extended-infusion dosing (22 [45.8%] vs. 2 [2.1%]; *p* < 0.001) were also more common among MDR-PSA cases, which received a shorter course of therapy (11.7 [95% CI: 9.6–13.9] versus 15.6 [95% CI 13.5–17.2]; *p* = 0.012).

### 3.4. Risk Factors for MDR-PSA Infection

In the univariate analysis, the following risk factors for MDR-PSA infection were identified: diabetes mellitus without target organ damage (OR: 2.45; 95% CI: 1.08–5.59; *p* = 0.033), receipt of antibiotic therapy within the previous 3 months (OR: 5.81; 95% CI: 2.53–13.33; *p* < 0.001), previous surveillance for MDR colonization (OR: 2.06; 95% CI: 1.02–4.16; *p* = 0.043), previous colonization by MDR bacteria (OR: 4.2; 95% CI: 1.81–9.74; *p* < 0.001) and MDR-PSA (OR: 23.5; 95% CI: 5.12–107.8; *p* < 0.001), and clinical presentation as septic shock (OR: 3.28; 95% CI: 1.49–7.21; *p* = 0.003).

In the multivariate model the presence of diabetes mellitus (adjusted OR [aOR]: 4.74; 95% CI: 1.63–13.79; *p* = 0.004), previous antibiotic therapy (aOR: 5.32, 95% CI: 1.93–14.73; *p* = 0.001), previous MDR-PSA colonization (aOR: 42.1; 95% CI: 4.49–394.8; *p* = 0.001), and septic shock (aOR: 3.73; 95% CI: 1.36–10.21; *p* = 0.010) emerged as independent risk factors (Table 3).

### 3.5. Complications and Outcomes

The occurrence of any type of complication during the index hospitalization was more common among MDR-PSA cases than non-MDR-PSA controls (14 [29.2%] vs. 11 [11.5%]; *p* = 0.008). In detail, *C. difficile* infection was more common in the former group (5 [10.4%] vs. 1 [1.0%]; *p* = 0.016). The rates of both overall (26 [54.2%] vs. 78 [81.3%]; *p* = 0.001) and 30-day clinical improvement (25 [52.1%] vs. 76 [79.2%]; *p* = 0.001) were significantly lower in MDR-PSA cases, as was the probability of being discharged alive by day 30 (17 [35.4%] vs. 61 [63.5%]; *p* = 0.001). On the other hand there were no significant differences in the length of hospital stay or the time to clinical improvement (Table 4).

## 4. Discussion

Infections due to MDR-PSA are an emerging threat for highly susceptible immunocompromised patients in a setting of global increase in multidrug resistance. Few studies, however, have been specifically focused on the clinical characteristics, predisposing factors and outcomes of MDR-PSA infection in this patient population [13,14]. The present case-control reveals that the presence of diabetes mellitus, the receipt of antibiotic treatment in the previous months, prior colonization by MDR-PSA, and the clinical presentation as septic shock should raise the suspicion of MDR-PSA involvement and prompt the early initiation of empirical antipseudomonal therapy in the immunocompromised host. Our experience also suggests a poorer outcome of episodes due to MDR-PSA isolates as compared to the non-MDR counterparts.

The independent impact observed for diabetes mellitus would point to a patient-related susceptibility to MDR-PSA infection beyond immune impairment. A recent meta-analysis has also reported that non-immunocompromised patients with type 2 diabetes are more prone to develop infections due to resistant bacteria in comparison to diabetes-free individuals [15]. Regarding previous MDR-PSA colonization and antibiotic therapy, both are well-established risk factors for MDR infections in the general population [16] and in specific types of immunocompromised patients —such as SOT [17] or HSCT recipients [18]—and once again emphasize the importance of assessing the patient’s history on an individual basis to inform the choice of antibiotic therapy. In fact, we observed significant differences between study groups in the appropriateness of empirical treatment, which was deemed adequate in less than a third of the MDR-PSA cases in contrast with more than two thirds of the non-MDR-PSA controls. Neither empirical combination therapy nor extended-infusion dosing regimens were commonly used (less than 50% and 20% of cases or controls, respectively), which could have negatively impacted on the probability of therapeutic success [19].

Septic shock at infection onset was identified as an independent predictor of MDR-PSA involvement, in line with previous studies reporting an association with higher severity of illness [20,21]. This finding would reflect a greater virulence of the MDR strains or an earlier clinical deterioration due to the delay in adequate treatment. We also found that the probability of clinical improvement or being discharged alive by day 30 since infection onset days was lower for the MDR-PSA group, even despite adequate matching for infection site, previous invasive procedures, major comorbidities and type of immunosuppression. This would support the hypothesis of a poorer prognosis associated to MDR-PSA [22]. Interestingly, it has been proposed that bacteremic SOT recipients under long-term immunosuppression may have a survival advantage over immunocompetent patients due to the modulation of the inflammatory response [23,24]. Patients with MDR-PSA infection also presented greater number of complications during the index episode of hospitalization, mainly driven by an increased risk of *C. difficile* infection.

The phenotypic profile of the included MDR-PSA isolates was representative of our institution during the study period, characterized by high resistance rates to carbapenems (25–45%) and aminoglycosides (20–45%) and VIM (Verona integron-borne metallo-β-lactamase) and GES (Guiana extended-spectrum β-lactamase) as the predominant carbapenemases in a high-endemicity setting for ST175 and ST235 high-risk clones [25]. Not unexpectedly, a sizeable proportion of imipenem-resistant isolates were still susceptible to meropenem (susceptibility rates of 6.3% and 22.9%, respectively), which would reflect the role of the loss of OprD or mutations in the promoter region of the *oprdD* gene [26]. It underlies the importance of knowing the local epidemiology to guide the design of empirical regimen.

The main strength of the present study is its case-control design with a 1:2 matching by relevant patient- and infection-related variables. Such an approach has been rarely applied in previous studies due to its complexity and the need of large numbers, and it was chosen in order to minimize residual confounding due to imbalances in baseline patient characteristics.

On the other hand, a number of limitations merit consideration. The single-center nature hampers the generalization of the results to different epidemiological scenarios. We have assembled a representative cohort of immunocompromised patients with PSA infection in daily practice, although the predominant syndrome by far was UTI (52.1% of all episodes), which also limits extrapolation to other more complex infections. In this line, only about one quarter of the patients had bacteremia. Newer antipseudomonal agents (such as ceftolozane-tazobactam or ceftazidime-avibactam) were not represented. Finally, the frequency of previous surveillance for MDR-GNB was unbalanced between cases and controls.

## 5. Conclusions

MDR-PSA infections in the immunocompromised host entails a poorer prognosis as compared to episodes due to non-MDR strains. The presence of patient-related (diabetes mellitus, previous receipt of antibiotics and MDR-PSA colonization in the preceding months) and infection-related factors (septic shock at infection onset) should prompt the initiation of adequate antipseudomonal empirical therapy in this population guided by local epidemiology at each center. The present experience supports the use of MDR colonization surveillance cultures for predicting the occurrence of MDR-PSA in immunocompromised patients.

## Figures and Tables

**Table 1 antibiotics-11-01459-t001:** Comparison of demographics and baseline patient characteristics of MDR-PSA cases and non-MDR-PSA controls.

Variable	MDR-PSA Cases (*n* = 48)	Non-MDR-PSA Controls (*n* = 96)	*p*
Age, years (mean ± SD)	62.8 ± 15.6	65.5 ± 14.1	0.295
Male gender, *n* (%)	32 (66.7)	73 (76.0)	0.23
Patient origin at admission, *n* (%)			0.707
Home	47 (97.9)	94 (97.9)	
Long-term care facility	0 (0.0)	2 (2.1)	
Other	1 (2.1)	0 (0.0)	
Hospital-acquired infection, *n* (%)	27 (40.9)	21 (26.9)	0.076
Charlson Comorbidity index, median (IQR)	3.5 (2–6)	3 (2–6)	0.98
McCabe-Jackson score, *n* (%)			0.31
Rapidly fatal (<3 months)	5 (10.4)	5 (5.2)	
Ultimately fatal (3 months to 5 years)	28 (58.3)	51 (53.1)	
Non-fatal (>5 years)	15 (31.25)	40 (41.7)	
Underlying disease, *n* (%)			
Diabetes mellitus	18 (13.3)	22 (22.92)	0.066
No target organ damage	15 (31.3)	15 (15.6)	0.03
Target organ damage	3 (6.3)	7 (7.3)	1.000
Chronic lung disease	3 (6.3)	10 (10.4)	0.544
Coronary heart disease	8 (16.7)	8 (8.3)	0.134
Other heart disease	5 (10.4)	10 (10.4)	1.000
Peripheral arterial disease	4 (8.3)	4 (4.2)	0.441
Cerebrovascular disease	2 (4.2)	2 (2.1)	0.601
Chronic kidney disease	18 (37.5)	36 (37.5)	1.000
Liver cirrhosis	4 (8.3)	4 (4.2)	0.441
HIV infection	2 (4.2)	3 (3.1)	1.000
Type of immunosuppression, *n* (%)			0.710
Solid organ transplantation	14 (29.2)	25 (26.0)	
Hematological malignancy	13 (27.1)	22 (22.9)	
Solid cancer	18 (37.5)	45 (46.9)	
Other ^a^	3 (6.3)	4 (4.2)	
Neutropenia, *n* (%)	9 (18.75)	15 (15.63)	0.635
Previous hospital admission, *n* (%) ^b^	18 (37.5)	26 (27.1)	0.201
Previous ICU admission, *n* (%) ^b^	2 (4.2)	3 (3.1)	1.000
Previous surgical intervention, *n* (%) ^b^	8 (16.7)	9 (9.4)	0.201
Previous receipt of antibiotics, *n* (%) ^c^	39 (81.3)	41 (42.7)	<0.001
Previous surveillance for MDR colonization, *n* (%) ^c^	26 (54.2)	35 (36.5)	0.043
Previous MDR colonization, *n* (%) ^c^	18 (37.5)	12 (12.5)	<0.001
Methicillin-resistant *Staphylococcus aureus*	1 (2.1)	1 (1.0)	1.000
ESBL-producing Enterobacterales	2 (4.2)	4 (4.2)	1.000
MDR *Klebsiella* spp.	2 (4.2)	2 (2.1)	0.601
MDR *Pseudomonas aeruginosa*	16 (33.3)	2 (2.1)	<0.001

ESBL: extended spectrum β-lactamase; HIV: human immunodeficiency virus; ICU: intensive care unit; IQR: interquartile range; MDR-PSA: multidrug-resistant *Pseudomonas aeruginosa;* OR: odds ratio; SD: standard deviation. ^a^ Includes rheumatic diseases under immunosuppressive treatment, advanced HIV infection, and other severe primary or secondary immunodeficiencies. ^b^ Within the 30 days preceding the index hospitalization. ^c^ Within the 3 months preceding the index hospitalization.

**Table 2 antibiotics-11-01459-t002:** Comparison of clinical characteristics of infectious episodes in MDR-PSA cases and non-MDR-PSA controls.

Variable	MDR-PSA Cases (*n* = 48)	Non-MDR-PSA Controls (*n* = 96)	*p*
Previous hospital stay, days (mean ± SD) ^a^	17.3 ± 34.5	9.8 ± 21.2	0.11
Invasive procedures or indwelling catheters in site, *n* (%)	27 (56.3)	46 (47.9)	0.346
Bladder catheter	22 (45.8)	35 (36.5)	
CVC	4 (21.1)	9 (25.7)	
Biliary tract prosthesis	1 (5.3)	2 (5.7)	
Site of infection, *n* (%)			1.000
Upper urinary tract infection	25 (52.1)	50 (52.1)	
Non-urinary catheter-related	12 (25.0)	22 (22.9)	
Urinary catheter-related	13 (27.1)	28 (29.2)	
Febrile neutropenia	7 (14.6)	14 (14.6)	
Non-ventilator-associated pneumonia	4 (8.3)	8 (8.3)	
Ventilator-associated pneumonia	1 (2.1)	2 (2.1)	
Primary bacteremia	4 (8.3)	8 (8.3)	
Intraabdominal infection	2 (4.2)	4 (4.2)	
Biliary tract infection	3 (6.3)	6 (6.3)	
Skin and soft tissue infection	1 (2.1)	2 (2.1)	
CVC-related bacteremia	1 (2.1)	2 (2.1)	
Overall bacteremia, *n* (%)	13 (27.1)	25 (26.0)	0.894
Clinical presentation, *n* (%)			
Sepsis	44 (91.7)	82 (85.4)	0.285
Septic shock	19 (39.6)	16 (16.7)	0.003
Pitt’s bacteremia score, *n* (%)			0.234
<2 points	24 (50.0)	58 (60.4)	
≥2 points	24 (50.0)	38 (39.6)	

CVC: central venous catheter; MDR-PSA: multidrug-resistant *Pseudomonas aeruginosa;* OR: odds ratio; SD: standard deviation. ^a^ Until diagnosis of PSA infection in the index hospitalization.

**Table 3 antibiotics-11-01459-t003:** Univariate and multivariate analysis of risk factors associated with MDR-PSA infection.

Variable	Univariate			Multivariate		
	OR	95% CI	*p*	aOR	95% CI	*p*
Diabetes mellitus with no target organ damage	2.45	1.08–5.59	0.033	4.74	1.63–13.79	0.004
Previous receipt of antibiotics	5.81	2.53–13.33	<0.001	5.32	1.93–14.73	0.001
Previous surveillance for MDR colonization	2.06	1.02–4.16	0.043	1.29	0.48–3.43	0.616
Previous MDR colonization	4.2	1.81–9.74	<0.001	0.29	0.05–1.64	0.161
Previous MDR *P. aeruginosa* colonization	23.5	5.12–107.8	<0.001	42.1	4.49–394.8	0.001
Septic shock at diagnosis	3.28	1.49–7.21	0.003	3.73	1.36–10.21	0.010

aOR: Adjusted Odds Ratio; CI: Confidence Interval; MDR: Multi-Drug Resistance; OR: Odds Ratio.

**Table 4 antibiotics-11-01459-t004:** Complications during the index hospitalization and outcomes in MDR-PSA cases and non-MDR-PSA controls.

Variable	MDR-PSA Cases (*n* = 48)	Non-MDR-PSA Controls (*n* = 96)	*p*
Any complication, *n* (%)	14 (29.2)	11 (11.5)	0.008
Secondary bacteremia	4 (8.3)	2 (2.1)	0.095
Requirement of invasive procedure	4 (8.3)	3 (3.1)	0.171
*Clostridioides difficile* infection	5 (10.4)	1 (1,0)	0.016
Digestive perforation	1 (2.8)	1 (1.0)	1.000
Septic thrombophlebitis	0 (0.0)	0 (0.0)	--
Secondary pneumonia	4 (8.33)	5 (5.21)	0.481
Other complications			
Secondary abscess due to *P. aeruginosa*	2 (4.2)	5 (5.2)	1.000
Thromboembolic or hemorrhagic event	2 (4.2)	6 (6.3)	0.719
Invasive fungal infection	0 (0.0)	3 (3.1)	0.551
Non-*Pseudomonas* secondary infection	3 (6.3)	3 (3.1)	0.4
Treatment-emergent adverse event, *n* (%)	3 (6.3)	5 (5.2)	1.000
Requirement of ICU admission during the index hospitalization, *n* (%)	14 (29.2)	22 (22.9)	0.414
Clinical improvement, *n* (%)	26 (54.2)	78 (81.3)	0.001
Time until clinical improvement, days (median [IQR])	11.5 (8–15)	12 (8–16)	0.701
30-day clinical improvement, *n* (%)	25 (52.1)	76 (79.2)	0.001
Discharged alive by day 30, *n* (%)	17 (35.4)	61 (63.5)	0.001
Length of hospital admission, days (mean [95% CI])	19 (12–27)	19 (15–23)	0.93

CI: confidence interval; ICU: intensive care unit; IQR: interquartile range; MDR-PSA: multidrug-resistant *Pseudomonas aeruginosa;* SD: standard deviation.

## Data Availability

The data presented in this study are available in the article.

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
