# Peer review of "Risk Factors and Outcomes for Multidrug Resistant Pseudomonas aeruginosa Infection in Immunocompromised Patients"

_antibiotics, 2022, doi:10.3390/antibiotics11111459_

Round 1

Reviewer 1 Report

1.     The results from the univariate and multivariate analysis shown in Table 1 and Table 2 are unorganized. But actually, the univariate and multivariate data do not match the purpose (figure title) of Table 1 and 2. Authors may put another table for the univariate and multivariate analysis data.

2.     Authors cited Ref18 to support their findings that septic shock is a predictor of MDR-PSA involvement. However, Ref18 does not mention that. Could the authors clarify this issue?

3.     In line 93, authors should provide the full names for VIM and GES.

Overall, this manuscript is well written, however, some issues should be addressed and clarified.

Author Response

Point 1. The results from the univariate and multivariate analysis shown in Table 1 and Table 2 are unorganized. But actually, the univariate and multivariate data do not match the purpose (figure title) of Table 1 and 2. Authors may put another table for the univariate and multivariate analysis data.

Response: We appreciate the point raised by the Reviewer. In response to his/her suggestion, we have modified and simplified Tables 1 and 2 to show only the comparison between both MDR-PSA cases and non-MDR-PSA controls. The order of paragraphs in the Results section has been accordingly modified to match those of variable depicted in tables. Finally, we have created a new Table 3 with the results of univariate and multivariate analysis.

Point 2. Authors cited reference #18 to support their findings that septic shock is a predictor of MDR-PSA involvement. However, reference #18 does not mention that. Could the authors clarify this issue?

Response: According to the reviewer suggestion, in reference #18 the authors reported that MDR Pseudomonas infection was associated with a higher severity index score, which implies a higher disease severity and would act as a surrogate for the presence of septic shock. We have modified the sentence in the Discussion section to avoid potential misunderstanding.

Point 3. In line 93, authors should provide the full names for VIM and GES.

Response: We appreciate the Reviewer's wording details. We modified the text according to these recommendations.

Reviewer 2 Report

The authors present results on an understudied infection. Although, the study is not large, it is focused on a specific population served by an academic medical center in a major metropolitan area. The approach is scientifically sound, well-described, and is appropriate to the level of conclusions. The statistical analysis is similarly appropriate. Given the increase in MDR pathogens and lack of previous work on PSA on this population, the work will be of general interest to the field and is appropriate for this journal.

Some minor concerns:

1) line 69 - I am unsure if this statement is sufficient. In some countries a specific authorization policy code/document/etc number must be provided. I am unfamiliar with the Spanish Government's requirements.

2) line 75-76 appears to be in a different font

Author Response

General comments. The authors present results on an understudied infection. Although, the study is not large, it is focused on a specific population served by an academic medical center in a major metropolitan area. The approach is scientifically sound, well-described, and is appropriate to the level of conclusions. The statistical analysis is similarly appropriate. Given the increase in MDR pathogens and lack of previous work on PSA on this population, the work will be of general interest to the field and is appropriate for this journal.

Response: We truly appreciate the positive comments raised by the Reviewer.

Point 1. Line 69: I am unsure if this statement is sufficient. In some countries a specific authorization policy code/document/etc number must be provided. I am unfamiliar with the Spanish Government's requirements.

Response: We have now confirmed in the revised manuscript that, according to the Spanish legislation, single-center retrospective studies only require the approval of the corresponding institutional clinical research ethics committee.

Point 2. Line 75-76 appears to be in a different font.

Response: We thank the Reviewer for raising this point. We have accordingly harmonized the style throughout the entire manuscript.

Reviewer 3 Report

I have reviewed the manuscript “Risk factors and outcomes for multidrug resistant Pseudomonas aeruginosa infection in immunocompromised patients.” submitted to “Antibiotics” for possible publication. In this paper, authors assessed the clinical characteristics, therapeutic management and outcomes in a single-center cohort of immunocompromised patients with MDR-PSA infection. In addition, in order to minimize potential confounding due to baseline conditions, authors analyzed the risk factors for MDR-PSA infection by means of a matched cohort composed of immunocompromised patients that also developed infection due to a PSA isolate with non-MDR phenotype. I found this work interesting, very nicely written and fit well within the scope of this journal. I don’t have any major comments, but the manuscript must need some minor improvements before acceptance; there are a few suggestions that authors need to consider in order to improve it further:

1.     Line 18-20: Revise the sentence.

2.     Line 20, 54, 57, 66: Rather than using we/us/our, authors are recommended to use current study/present study.

3.     Replace “frailty” with “weakness”.

4.     Line 39: The in-text citations for reference needs to be corrected. This is not the proper way of citing a reference.

5.     Authors needs to write more paragraph in the introduction section with more references.

6.     Line 129: The authors mentioned that “Intrinsic resistance was not considered”. Here author should clarify that either they did not consider intrinsic resistance for MDR category or what? If for intrinsic resistant, then remove this sentence as it is already well known.

7.     Page 8, Line 2-8: Authors are advised to revise the sentence as something is looking wrong. At line 2 authors mentioned that the non-susceptibility against carbapenems was 77.1%, while in line 7 they mentioned that susceptibility to imipenem was <7%.

8.     Did author wanted to say non-susceptibility as resistant? Please clarify.

9.     Page 10, Line 111: Authors can write conclusion as separate heading.

Author Response

Point 1. Line 18-20: Revise the sentence.

Response: According to the Reviewer suggestion, the sentence has been reworded as follows: “A deeper knowledge of the risk factors for MDR-PSA infection in this patient population may help to choose appropriate empirical antibiotic therapy”.

Point 2. Lines 20, 54, 57, 66: Rather than using we/us/our, authors are recommended to use current study/present study.

Response: The manuscript has been modified in response to the Reviewer suggestion.

Point 3. Replace “frailty” with “weakness”.

Response: We have modified this in the revised text.

Point 4. Line 39: The in-text citations for reference needs to be corrected. This is not the proper way of citing a reference.

Response: The manuscript has been accordingly chaged.

Point 5. Authors needs to write more paragraph in the introduction section with more references.

Response: According to the reviewer suggestions, we have enlarged the Introduction section and added two more references:  

Falagas, M.E.; Kopterides, P. Risk factors for the isolation of multi-drug-resistant Acinetobacter baumannii and Pseudomonas aeruginosa: a systematic review of the literature. J Hosp Infect. 2006 Sep;64(1):7-15.

Kara Ali, R.; Surme, S.; Balkan, I.I.; Salihoglu, A.; Sahin Ozdemir, M.; Ozdemir, Y., et al. An eleven-year cohort of blood-stream infections in 552 febrile neutropenic patients: resistance profiles of Gram-negative bacteria as a predictor of mor-tality. Ann Hematol. 2020 Aug;99(8):1925-1932.

Point 6. Line 129: The authors mentioned that “Intrinsic resistance was not considered”. Here author should clarify that either they did not consider intrinsic resistance for MDR category or what? If for intrinsic resistant, then remove this sentence as it is already well known.

Response: The sentence has been accordingly reworded to clarify that intrinsic resistance was not considered in the definition of MDR, in line with the commonly used criteria proposed by Magiorakos et al (Clin Microbiol Infect 2012; 18: 268-81).

Point 7. Page 8, line 2-8: Authors are advised to revise the sentence as something is looking wrong. At line 2 authors mentioned that the non-susceptibility against carbapenems was 77.1%, while in line 7 they mentioned that susceptibility to imipenem was <7%.

Response: The sentence pointed by the Reviewer is referred to "non-susceptibility" (i.e. resistance) to any carbapenem, which was indeed over 75%. Afterwards in the paragraph we specifically detailed that the susceptibility rate to imipenem alone was lower than 7%, which implied that a sizeable proportion of isolates exhibited resistance to imipenem but not meropenem. This finding is not completely unexpected, since imipenem resistance in Pseudomonas aeruginosa is typically associated with loss of the porin OprD or mutations in the corresponding gene (often combined with overproduction of chromosomal AmpC beta-lactamase). We have addressed this point in the Discussion section of the revised manuscript (page 11).

Point 8. Did author wanted to say non-susceptibility as resistant? Please clarify.

Response: Effectively, we use the term “non-susceptibility” as synonymous of “resistance”. According to the Reviewer suggestion, and to avoid any potential misunderstanding, we have harmonized this terminology throughout the paper.

Point 9: Page 10, line 111: Authors can write conclusion as separate heading.

Response: The manuscript has been accordingly changed.

Round 2

Reviewer 1 Report

All my concerns are addressed. There are just some minor issues in the manuscript.

In line 15 there is a typo "(Table 2)="

In line 149, Núñez, JA.. -->two dots.